# Understanding How Intelligence and Academic Underachievement Relate to Life Satisfaction Among Adolescents with and Without a Migration Background

**DOI:** 10.3390/jintelligence13090105

**Published:** 2025-08-22

**Authors:** Alicia Neumann, Ricarda Steinmayr, Marcus Roth, Tobias Altmann

**Affiliations:** 1Department of Psychology, University of Duisburg-Essen, 45141 Essen, Germany; 2Department of Psychology, TU Dortmund University, 44227 Dortmund, Germany

**Keywords:** intelligence, achievement, underachievement, life satisfaction, adolescence, migration background

## Abstract

Intelligence, academic achievement and an unfavorable discrepancy between them (i.e., underachievement) have been proposed to influence students’ subjective well-being. However, previous research on these effects remains scarce and inconsistent. The present study examined the associations between said variables in a sample of 695 fifteen-year-old students in Germany, differentiating between those with and without a migration background. Our findings unexpectedly revealed that students with a migration background reported higher life satisfaction than those without a migration background. Intelligence was unrelated to life satisfaction, regardless of migration background. Academic achievement, measured by the grade point average, was positively associated with life satisfaction among students without a migration background but showed no such relationship in students with a migration background. Segmented regression analyses further indicated that an unfavorable discrepancy between IQ and grade point average, reflecting underachievement, was associated with lower life satisfaction among students with a migration background but not among those without. These findings partially challenge previous research and theoretical assumptions. We discuss the theoretical and practical implications of our findings for educational policy and emphasize the importance of targeted interventions to address underachievement in students with a migration background. Our findings suggest that poor academic performance can have a particular impact on well-being in this group. Accordingly, interventions aimed at reducing the achievement gap of those students should not only target cognitive and academic skills but also promote emotional support, cultural inclusion and social integration in the school environment.

## 1. Introduction

Understanding how cognitive abilities and academic achievement contribute to students’ subjective well-being (SWB) has been a longstanding focus in psychological research. While SWB has been widely studied in relation to intelligence, academic performance, and educational outcomes, not only the strength but even the direction of these presumed associations remains unclear. While numerous studies have found positive associations between intelligence, academic achievement, and SWB (e.g., [9]; [26]), other studies have found these associations to be weak or inconsistent (e.g., [19]; [57]). In addition, students who perform below the expected academic level—underachievers—may have lower levels of well-being ([55]). This is in line with findings from [49] ([49]) who showed that students’ grade point average positively predicts changes in life satisfaction.

Despite the growing interest in this area, the underlying mechanisms linking intelligence, underachievement and SWB remain poorly understood, particularly in the context of sociocultural diversity. Recent evidence suggests that underachievement is more prevalent among students with a migration background (e.g., [22]; [45]), highlighting the need to examine group-specific patterns.

The present study aims to make a contribution to the above-mentioned uncertainties by examining how intelligence and academic underachievement affect the life satisfaction of students with and without a migration background.

### 1.1. SWB and Life Satisfaction

According to [15] ([15]), SWB encompasses both cognitive and affective facets. Affect encompasses negative and positive moods and emotions. Affective well-being specifically means that positive emotions are present and negative emotions are absent.

[56] ([56]) postulates that the term life satisfaction is predominantly used in the sense of a comprehensive feeling of overall SWB, but in certain contexts it refers specifically to the cognitive evaluation of one’s own life and is then synonymous with contentment.

Diener states that “subjective well-being is the scientific term for happiness and life satisfaction” ([16]). In any case, there is a strong convergence between the affective and cognitive component of SWB, which [5] ([5]) indicate with a correlation of 0.78 and 0.91. Based on these findings, we assume that the cognitive aspect of SWB is sufficient as a proxy to draw general conclusions about SWB. To ensure transparency, however, the term SWB is used in the paper when emotional and cognitive aspects are involved and the term life satisfaction when only cognitive aspects are in focus.

### 1.2. Intelligence

General intelligence can be seen as a multifaceted cognitive capacity that encompasses a range of abilities. These include the ability to analyze critically, solve problems or understand and process complex information and ideas ([18]; [33]). It correlates significantly with mathematical understanding, verbal competence and spatial reasoning ([60]). According to [44] ([44]) intelligence is often divided into two categories: On the one hand, there is crystallized intelligence, which is related to knowledge acquired from the environment or culture, and fluid intelligence, i.e., the ability to solve new problems on the basis of logical thinking (without relying on acquired knowledge).

### 1.3. The Relations Between Intelligence and SWB

Previous studies on the links between intelligence and SWB often report positive associations; however, they do not do this consistently. On the one hand, intelligence has been argued to serve as a protective factor against the early development of mental health problems and reduced well-being because of associated abilities such as recognizing stressors and applying effective problem-solving strategies to cope with stress and challenges ([17]), allowing more intelligent individuals to navigate through obstacles and setbacks more successfully ([8]; [32]). Accordingly, intelligence was found to be negatively associated with psychological distress ([17]) and positively associated with SWB ([35]). According to [44] ([44]), only the construct of fluid intelligence is positively associated with life satisfaction in a group of younger adults, but not crystallized intelligence.

On the other hand, however, there are also several studies that report nonsignificant associations. In their analysis of 23 studies taken from the World Database of Happiness, [57] ([57]) found no significant relationship between intelligence and SWB at the individual level, indicating that higher intelligence does not necessarily correspond to higher levels of SWB. Similarly, [19] ([19]) concluded that general intelligence is not linked to SWB levels. In trying to explain these findings, it has been argued that individuals with higher intelligence may have elevated and unattainable expectations which leads to dissatisfaction and thus eradicates the otherwise positive association between intelligence and SWB ([57]).

Due to recent societal changes in many countries, involving increasing rates of international migration, several studies also explored the relevance of a migration background in these aforementioned associations. Most studies indicate that students with a migration background report lower SWB, lower school-related well-being, higher prevalence rates of anxiety, and show lower academic achievement compared to their non-migrant peers ([27]; [29]; [34]; [36]; [53]). This pattern was also supported by findings from the 2015 PISA study ([37]). Previous PISA studies further indicate that migrant adolescents consistently score lower across various academic performance measures compared to their non-migrant peers ([4]; [59]).

Regarding the associations between intelligence and SWB in students with a migration background, previous studies remain inconclusive. Some studies suggest that the relationship between intelligence and SWB is weaker among individuals with a migration background than among those without (e.g., [1]). This may be attributed to additional challenges and stressors that migrants face in the foreign country which may negatively impact their well-being ([57]). However, several other studies report that migration background is not significantly associated with SWB at all (e.g., [24]; [52]).

### 1.4. The Relations Between Achievement and SWB

Similarly to the links between intelligence and SWB as detailed above, the links between academic achievement and SWB have also been widely studied and debated in the previous literature on how these variables influence each other. Academic achievement (mostly operationalized by grades or grade average) is generally assumed to positively affect the various facets of SWB which is supported by previous studies both cross-sectionally and longitudinally ([9]; [26]; [49]). A meta-analysis by [10] ([10]) comparing low- to high-achieving students showed that the correlation between academic achievement and SWB was small to medium in magnitude and statistically significant. This relationship was also shown to be reciprocal in nature: Students with higher SWB were found to be more likely to achieve better final grades, and higher academic success positively influenced students’ SWB ([38]). Thus, SWB can function both as a prerequisite for and a consequence of strong academic performance, suggesting that students may perform well in school because they are generally satisfied with their lives, while good grades contribute to their overall satisfaction ([14]; [34]). Conversely, low academic performance may increase anxiety, worry, and stress ([27]; [53]) and may cause students to experience high pressure to perform well which may ultimately reduce their SWB ([46]). It should be noted that despite this broad consensus, several of these studies report the effect sizes to be low ([25]). According to the 2015 PISA study, the correlation between school performance and life satisfaction is weak, with students at high and low performance levels reporting similar levels of life satisfaction in most countries ([37]).

Based on these findings, it has been argued that not the absolute level of achievement relates to SWB, but the level of achievement measured against the individual capacity of achievement. The negative discrepancy between cognitive ability (intelligence) and academic achievement has been termed underachievement ([40]). Previous studies linking underachievement to SWB supported these discrepancy assumptions and reported an association between low well-being and underachievement in students in general ([55]) and in gifted students in particular ([11]). However, these previous studies are scarce so that replications are needed to validate the reliability of the findings.

The individual level of underachievement has also been shown to co-vary with migration background. Although underachievement is most prominently found among students from disadvantaged and low-educated familial backgrounds, substantial differences between students with and without a migration background have been reported as well, with underachievement found more prominently in students with a migration background ([7]). Several attempts have been made to explain these results, for instance, by controlling for differences in socio-economic status. However, even then both the achievement gap and the differences in underachievement between migrant and native students remain substantial ([2]; [39]). Thus, migration background needs to be considered a relevant factor in explaining differences in achievement, underachievement, and SWB.

### 1.5. The Present Study

As outlined at the beginning, the research results on the relationship between intelligence, academic performance, underachievement and SWB are characterized by inconsistencies. While a number of studies have found positive associations between intelligence, academic achievement and SWB (e.g., [9]; [26]), other research has indicated weak or inconsistent links between the constructs (e.g., [19]; [57]). The sometimes contradictory findings and the overall limited number of empirical studies make a well-founded assessment of this correlation difficult. Furthermore, empirical studies rarely examine how these relationships differ between students with and without a migration background. The present study closes this gap by jointly analyzing the effects of intelligence, academic achievement, underachievement and migration background on life satisfaction in a large sample of adolescents.

Based on the aforementioned theoretical considerations, we phrased the following hypotheses. H1: Intelligence and SWB are positively related in both students with and without a migration background. H2: Academic achievement and SWB are positively related in both students with and without a migration background. H3: Students with a migration background are more likely to show an unfavorable intelligence-grade discrepancy (underachievement) compared to students without a migration background. H4: The discrepancy between intelligence and academic achievement is associated with life satisfaction such that a less favorable discrepancy (underachievement) is linked to lower life satisfaction. Differences in these relations between students with and without a migration background were treated exploratorily, without a specific hypothesis being put forth.

## 2. Materials and Methods

### 2.1. Data Collection and Study Design

The study was preregistered: https://doi.org/10.17605/OSF.IO/JQE57 (accessed on 10 January 2025.

The dataset originated from the study “FA(IR)BOLOUS” ([48]), which aimed to investigate social inequalities in school transitions. The research sought to determine whether educational disadvantages based on socioeconomic background diminish or intensify during the transition to secondary school. The study surveyed ninth-grade students from 15 secondary schools in Germany. Participation was voluntary. Schools were randomly selected to ensure a socially heterogeneous and regionally representative sample. The survey was conducted in a classroom setting, anonymously, voluntarily, and under the supervision of two trained test administrators.

### 2.2. Sample

723 participants took part in the study. Of these, 28 were excluded due to highly implausible response patterns, suggesting a lack of honest participation (e.g., consistently selecting only the maximum or the minimum response option across all items of the questionnaire regardless of inverted items). The final sample consisted of 695 students, with a mean age of 15.1 years (*SD* = 0.7). They ranged in age from 13 years to 17 years. Within this sample, 48.9% were female, and 54.0% reported a migration background. The relatively high proportion of students with a migration background can be attributed to two factors: First, the sample was primarily drawn from the Rhine-Ruhr region in Germany which represents a particularly diverse area in Germany, and second, the study included students from comprehensive and middle-track secondary schools (i.e., “Gesamtschule” and “Realschule”) where the proportion of students with a migration background is considerably larger than in upper-track secondary schools (i.e., Gymnasium) ([47]).

### 2.3. Measures

#### 2.3.1. Sociodemographic Data

Using a sociodemographic questionnaire, we collected information on participants’ age, gender, and migration background. To determine the existence of a migration background, participants were asked three questions: (1) Were you born outside of Germany? (2) Was your mother born outside of Germany? (3) Was your father born outside of Germany? If participants answered “yes” to any of these questions, the participant was coded as “with migration background.”

#### 2.3.2. Intelligence

Intelligence was assessed using the Culture Fair Test CFT 20-R by [61] ([61]), a widely applicable measure of fluid intelligence based on figural task material. This test is designed for individuals aged 8 to 64 years and consists of four task categories: continue figure series, classify figures, complete figure matrices, and draw topological inferences. As the CFT-20 R works with language-free test items, people with little knowledge of German and insufficient cultural skills are not disadvantaged. As a large proportion of our sample has a migrant background, we were able to ensure that no one was treated unfairly by using this test. The CFT 20-R was carried out as a group test by trained research personnel. The short version of the test was used for the study.

In our sample, the mean IQ was 98.3 (*SD* = 14.7). The test demonstrated good internal consistency with a Cronbach’s alpha of 0.77 and a split-half reliability of 0.68 (*p* < .001).

#### 2.3.3. Academic Achievement

Academic achievement was assessed using school grades as provided by the teachers on the students’ report cards. In the German school system, grades range from 1 to 6, with 1 representing the highest achievement and 6 the lowest. For the presentation of average scores, we retained this original scaling (higher scores indicating lower achievement). However, for all correlational analyses, we inverted the scaling to facilitate interpretation (higher scores indicating higher achievement). To represent the overall academic achievement, we calculated the average grade across the three core subjects German (first language), mathematics, and English (first foreign language). This selection is justified theoretically by the central role of these subjects in the German school system curriculum as well as empirically by their substantial intercorrelations which ranged from 0.41 to 0.48 (all *p*s < .001).

#### 2.3.4. Underachievement/Overachievement

To create a measure of underachievement which captures the unfavorable discrepancy between cognitive potential and actual academic performance, we computed the IQ-grade discrepancy. This required to initially transform both IQ and grades into standardized and bounded variables within the same domain to ensure comparability. To achieve this, we calculated percentile ranks for both IQ and grades separately, which per definition range from 0% to 100%. In these percentile rank variables, a given percentage represents the proportion of participants who achieved the same or a better score. For instance, an IQ percentile rank of 2% indicates that 2% of the sample obtained the same or a higher IQ score than the participant with this rank. To facilitate interpretation, we inverted the grade variable so that higher scores indicate higher achievement to parallel the IQ rank. Thus, a percentile rank of 10% in the inverted grade variable means that 10% of the sample achieved the same or a better grade. We then computed the IQ-grade discrepancy as the difference between the two rank variables (IQ and grade), resulting in a score that could range from −100 to 100. A score of −100 means that the participant had the highest IQ and the lowest grade in our sample (underachievement). A score of 0 signifies that IQ and grade percentile ranks were identical, reflecting zero underachievement. A score of 100 represents a participant who had the lowest IQ but the highest grade in our sample (which could be termed overachievement). This IQ-grade discrepancy score serves as a measure of underachievement (below zero) and overachievement (above zero), capturing the discrepancy between cognitive potential and actual academic performance.

This procedure is based on the simple difference method (e.g., [28]). If the expected level of achievement (based on ability) is compared with the actual level of achievement (e.g., grades or test results) and the difference is at least one standard deviation, this is referred to as underachievement. In their study comparing different methods for identifying underachievement, [23] ([23]) conclude that the strongest evidence for the validity of an identification method is found in the simple difference method. They therefore describe this method as the ideal method for identifying underachievement. We use this approach and treat the resulting difference variable as a continuous variable instead of categorizing it to best preserve its information.

#### 2.3.5. Life Satisfaction

To measure SWB, we used the seven-items Life Satisfaction subscale (GLS) from the Habitual Subjective Well-Being Scale ([13]). A sample item reads, “I am satisfied with my life situation.” Responses were made on a six-point rating scale ranging from 1 (“That’s not true at all”) to 7 (“That’s exactly right”). The subscale showed high internal consistency with Cronbach’s alpha = 0.89.

The Habitual Subjective Well-Being Scale consists of the GLS, which measures the cognitive-evaluative dimension of SWB, and the Mood Level Scale, which measures the affective dimension. We used only the seven items of the GLS because this was our main concern and we consider life satisfaction as an indicator of SWB, as explained above. In empirical studies examining children’s and adolescents’ SWB, life satisfaction is the most commonly used indicator (see, e.g., [38]; [42]; [51]).

### 2.4. Data Analysis

First, we calculated the descriptive statistics and mean differences between students with and without a migration background. Second, we tested for bivariate correlations in students without and with a migration background. Third, we employed a segmented regression approach, which fits well to the concept of underachievement. As described above, underachievement can be represented by an IQ-grade discrepancy score below zero and overachievement can be represented by an IQ-grade discrepancy score above zero. Although measured on a continuous scale, underachievement and overachievement may differ in their associations with life satisfaction, as described above. In this case, the regression line for underachievement and the regression line for overachievement should diverge, indicating a breakpoint between the regression lines around an IQ-grade discrepancy score of zero. This exploratory analysis was conducted using the R package “segmented” version 2.1 ([31]). Correlation coefficients below and above the break point were computed for the total sample and separately for students without and with a migration background.

## 3. Results

### 3.1. Mean Differences and Bivariate Correlations

First, we provide the descriptive statistics and mean differences between students with and without a migration background. Students with a migration background reported higher life satisfaction (*M* = 5.2, *SD* = 1.1) compared to those without a migration background (*M* = 4.8, *SD* = 1.2), with a medium effect size (*t*(df = 693) = 4.59, *p* < .001, *d* = 0.35). Contrary to our third hypothesis, the IQ-grade discrepancy, used as an indicator of underachievement/overachievement, did not differ significantly between the two groups (without migration background: *M* = 6.8, *SD* = 35.3; with migration background: *M* = 7.3, *SD* = 34.0; *t*(df = 693) = 0.18, *p* = .859, *d* = 0.01). However, the grade average differed significantly (*t*(df = 693) = 2.47, *p* = .014, *d* = 0.19), with students without a migration background achieving a lower (i.e., better) grade average (*M* = 3.1, *SD* = 0.7) compared to students with a migration background (*M* = 3.2, *SD* = 0.7), though the effect size was small.

Second, we tested for bivariate correlations in students without and with migration background. Recall that grades here are used in their inverted scaling with lower scores indicating lower achievement and higher scores indicating higher achievement (see Matherials and Methods Section 2). Grade average was positively related to IQ in both students with (*r* = 0.31, 95% CI [0.21, 0.40], *p* < .001) and without a migration background (*r* = 0.21, 95% CI [0.11, 0.32], *p* < .001). Contrary to our first hypothesis, life satisfaction was unrelated to IQ in either group (|*r*| < 0.06, *p* > .236). Life satisfaction showed a weak positive association with the grade average in students without a migration background (*r* = 0.11, 95% CI [0.00, 0.22], *p* = .048), indicating that better grades were linked to higher satisfaction in this group. However, contrary to our second hypothesis, this relationship was not significant for students with a migration background (*r* = 0.08, 95% CI [−0.02, 0.18], *p* = .103). Lastly, while the correlation between IQ-grade discrepancy and life satisfaction was not significant for students without a migration background (*r* = 0.05, *p* = .402), it was significant for those with a migration background (*r* = 0.11, *p* = .039).

Table 1 displays the correlations for all study variables, including means and standard deviations.

### 3.2. Segmented Regression Analysis

To further explore these relationships, we employed a segmented regression approach as described above. The assumptions of regression analyses regarding homoscedasticity (Breusch-Pagan test BP = 0.836, *p* = .361) and normally distributed residuals (visual inspection of the Q–Q plot) were met. To test this, we used the R package “segmented” ([31]), which identified a break point in the IQ-grade discrepancy variable at 13.6 (*SE* = 23.2), as shown in Figure 1. Below this break point, the association between IQ-grade discrepancy and life satisfaction was positive (β = 0.0052, 95% CI [0.0003, 0.0101], b = 0.0052, SE = 0.0025, t = 2.09, p = .0366), indicating that as the IQ-grade discrepancy increases from its lowest value (−100) to the midpoint (0), life satisfaction scores increase by 0.5 points. Considering that life satisfaction was measured on a 1–7 scale, this change corresponds to an 8.3% increase. Above the break point, however, IQ-grade discrepancy and life satisfaction were unrelated (β = −0.0014, 95% CI [−0.0089, 0.0062], b = 0.0066, SE = 0.0046, t = −1.43, n.s.), suggesting that once IQ and grade are approximately aligned (i.e., percentile ranks of IQ and grade are approximately equal), further increases in IQ-grade discrepancy do not lead to additional gains in life satisfaction. Figure 1 illustrates the segmented regression with 95% confidence intervals.

We then computed correlation coefficients below and above the break point for the total sample and differentiating between students without and with migration background. Using the total sample, the correlation was significant below the break point (*r* = 0.10, *p* = .042, *n* = 401) but not above it (*r* = −0.02, *p* = .679, *n* = 294). However, the difference between these correlation coefficients was non-significant (Steiger’s *z* = 1.56, *p* = .059) by a small margin. Interestingly, the correlation coefficients below and above the break point did not differ and were both nonsignificant in students without migration background (below: *r* = 0.03, *p* = .722, *n* = 175; above: *r* = 0.00, *p* = .964, *n* = 145; *z* = 0.27, *p* = .396). However, in students with migration background the correlation below the break point was significant (*r* = 0.15, *p* = .021, *n* = 226), the correlation above the break point was not (*r* = −0.08, *p* = .325, *n* = 149), and these two coefficients differed significantly (*z* = 2.17, *p* = .015). The positive correlation below the break point indicates that higher levels of underachievement are associated with lower levels of life satisfaction (see Figure 1). Thus, migrant students with high IQ ranks and poor grade ranks tend to report lower levels of life satisfaction. This relation only holds up until the breakpoint. Above the breakpoint, a balanced or even favorable IQ-grade discrepancy does not contribute to higher levels of life satisfaction.

## 4. Discussion

The present study examined the complex interplay between intelligence, academic achievement, underachievement, and migration background in relation to the life satisfaction of ninth-grade students in Germany. Several findings deviated from our expectations and, consequently, from prevailing theoretical assumptions regarding these associations.

Our first hypothesis was that intelligence and SWB are positively related in both students with and without a migration background. As mentioned at the beginning, there have been some studies that have not found a positive relationship between intelligence and SWB. The inconsistent findings led us to investigate this question and our findings also showed that there is no correlation between the two constructs in the given sample. This applied regardless of migration status. [57] ([57]) demonstrated that individuals with higher intelligence do not necessarily experience greater SWB. Similarly, [19] ([19]) found no association between intelligence and SWB. One possible explanation is that individuals with higher intelligence often have elevated expectations, which may not always be fulfilled ([57]).

Our second hypothesis was that academic achievement and SWB are positively related in both students with and without a migration background. The results partially supported our hypothesis and showed that academic success was positively associated with life satisfaction among students without a migration background but not among those with a migration background. This suggests that sociocultural factors may play a role in the relationship between academic achievement and SWB.

Differences in value systems and expectations could account for this pattern. Specifically, academic success may be more closely linked to life satisfaction for students without a migration background, as their families and social environments could emphasize academic achievement as a key determinant to personal fulfilment.

Moreover, prior research suggests that social and familial support can buffer the negative impact of academic setbacks on well-being ([41]). Students with a migration background may derive their sense of SWB and life satisfaction more from family and community support than from individual academic success. Additionally, experiences of discrimination or lower teacher support may lead these students to feel less academic achievement and reduced enjoyment in school, thereby weakening the link between grades and life satisfaction ([43]). Given that students with a migration background are more likely to encounter such challenges, this could explain the weaker association observed in our study.

We also hypothesized that students with a migration background would be more likely to exhibit an unfavorable IQ-grade discrepancy, given the additional challenges they face and their increased risk of underachievement. However, this assumption was not supported by our findings.

Our fourth hypothesis stated that the discrepancy between intelligence and academic achievement is associated with life satisfaction such that a less favorable discrepancy (i.e., underachievement) is linked to lower life satisfaction. Our results revealed that while the relationship between IQ-grade discrepancy and life satisfaction was not significant for students without a migration background, it was significant for those with a migration background. Segmented regression analysis revealed that students with an unfavorable IQ-grade discrepancy tend to report lower levels of life satisfaction. This relation only holds up until a breakpoint. Above this breakpoint, a balanced or even favorable IQ-grade discrepancy does not contribute to higher levels of life satisfaction.

A very interesting finding from our study was that students with a migration background reported higher life satisfaction than those without, a result that contrasts with many previous studies ([30]; [54]; [58]). This is an interesting finding in light of the fact that it is often assumed that young people with a migration background tend to be less satisfied due to social disadvantages, language barriers or discrimination. However, our results show the opposite and indicate that life satisfaction is not solely determined by external disadvantages. This raises the question of whether migration background has become less significant in Germany, particularly in diverse regions such as the Ruhr area, or whether its influence on life satisfaction is less pronounced than previously assumed. [12] ([12]), using data from the BaM (Becoming a Minority) project, found that an increasing number of European cities are becoming “majority-minority cities,” where over 50% of the population has a migration background. Their study suggested that residents of such neighborhoods (e.g., Amsterdam, Rotterdam, Malmö, and Vienna) often do not perceive themselves as belonging to a minority group. This shift in perception may mitigate many of the negative psychological effects typically associated with minority status. A similar mechanism may be at play in our study, as the proportion of students with and without a migration background was roughly equal. The perception of not being a minority, despite being classified as having a migration background, could partially explain our why students with a migration background were more satisfied with their life and why they were not more likely to exhibit an unfavorable IQ-grade discrepancy compared to native students.

### 4.1. Practical Implications

Our study revealed that an unfavorable IQ-grade discrepancy (underachievement) was associated with life satisfaction among students with a migration background but not among native students. Low academic achievement remains a significant challenge in modern educational systems, hindering institutions from effectively fulfilling their educational mission. Struggling students may disrupt the learning environment, potentially affecting both teachers and peers ([3]). Given these challenges, it is crucial to gain deeper insights into low academic achievement and its associations with psychological constructs. Our findings suggest that the impact of an unfavorable IQ-grade discrepancy on the life satisfaction of students with a migration background warrants further research and targeted interventions. Schools and educators should focus on this vulnerable group, ensuring that their specific needs are recognized and addressed.

If we consider the result, we can draw some interesting conclusions. In particular, the finding that students without a migration background have a slightly but significantly better average grade than students with a migration background is very interesting given the fact that students with a migration background report higher life satisfaction than those without. Our findings thus underline that overall intelligence and life satisfaction are not related. Schools could learn from this, for example, that life satisfaction may not only be promoted by academic success (in the form of good grades), but also by other influences such as emotional support, cultural recognition, teacher-student-relationship ([6]) and social integration.

### 4.2. Limitations

This study has several limitations. As it relies on cross-sectional data, the extent to which the findings can be generalized to other age groups, cohorts, etc., remains unclear. A longitudinal study tracking adolescents over time could address this limitation and provide valuable insights.

Additionally, data collection occurred before the COVID-19 pandemic, which may limit the generalizability of the findings to more recent cohorts. Furthermore, the measurement of migration background requires further scrutiny. “Migration background” is a highly heterogeneous variable, and in this study, we did not differentiate between first- and second-generation immigrants. However, [50] ([50]) demonstrated that these groups can have distinct experiences. Their findings in an adolescent sample indicated that first-generation immigrants were at a higher risk of experiencing bullying and physical altercations, whereas this risk decreased among second-generation immigrants. Future research should account for these distinctions and further explore the diverse ethnocultural backgrounds within the sample of students with a migration background. Above that, our measurement of intelligence deserves attention. While fluid intelligence tends to have a very high impact on general intelligence (e.g., [20], [21]), the two constructs are not completely equivalent. Rather, general intelligence is a conglomerate of different cognitive abilities and cannot be reduced to a single broad factor. It is unclear to what extent our results would have changed if we had considered other measures of intelligence. Further research should address this issue.

## Figures and Tables

**Figure 1 jintelligence-13-00105-f001:**
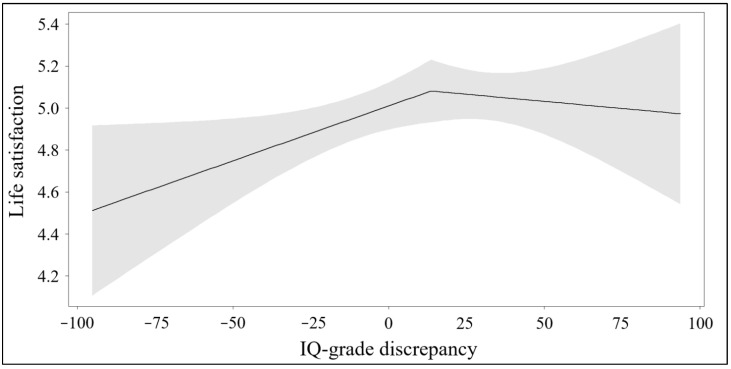
Segmented regression model of the relationship between life satisfaction and IQ-grade discrepancy with two regression lines, before and after the break point. Shaped areas indicate the 95% confidence intervals.

**Table 1 jintelligence-13-00105-t001:** Descriptive Statistics and Bivariate Correlations of Study Variables.

	Descriptive Statistics			Correlation Coefficients		
	Theoretical Range	Empirical Range	*M* (*SD*)	IQ	Grade Average	IQ-Grade Discrepancy
Life satisfaction	1.0; 7.0	1.1; 7.0	5.0 (1.2)	−0.04	−0.08 *	0.08 *
Intelligence	-	57.0; 148.0	98.1 (14.1)		−0.27 ***	−0.59 ***
Grade average (not inverted)	1.0; 6.0	1.0; 5.7	3.1 (0.7)			−0.57 ***
IQ-grade discrepancy	−100.0; 100.0	−95.3; 93.8	7.0 (34.6)			

Note. * *p* < 0.05, *** *p* < 0.001.

## Data Availability

Data are available from the corresponding author upon request.

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
