# Peer review of "Understanding How Intelligence and Academic Underachievement Relate to Life Satisfaction Among Adolescents with and Without a Migration Background"

_jintelligence, 2025, doi:10.3390/jintelligence13090105_

Round 1

Reviewer 1 Report

Comments and Suggestions for Authors

Thank you for the opportunity to review the article entitled Understanding How Intelligence and Academic Underachievement Relate to Life Satisfaction Among Adolescents With and Without a Migration Background

The primary aim of this research was to examine the relationships between intelligence, academic achievement, underachievement and students' subjective well-being, differentiating between students with and without a migration background.

I have a few comments on the manuscript, which are outlined below.

Comment 1

- The introduction provides a strong theoretical foundation by citing relevant literature.

Comment 2

Materials and Methods

Point 2.3.3.

The authors explain the procedure they have followed to identify underachievers, but do not provide any citations to support their choice. There are many papers on the subject of underachiever identification, in which commonly used identification methods are presented and defined.

Authors are requested to summarize and cite commonly used methods of underachiever identification (with supporting citations), explain why they have used the method selected in this study (with supporting citations), and what benefits the selected method has over commonly used methods.

Comment 3

Results

- In accordance with APA guidelines, please include the full table of correlations for all study variables, including means and standard deviations.

- Please comment on whether the analysis of regression assumptions has been carried out and indicate whether they have been met.

The article makes a relevant contribution to the advancement of knowledge. It is well written, the methodology is correct and well applied. Therefore, I consider the article to be of quality and, if the indicated modifications are included, it can be accepted for publication.

Reviewer 2 Report

Comments and Suggestions for Authors

Thank you for the opportunity to review this paper. Please find below a list of comments pointing to specific aspects of the paper that need some kind of revision. Most of these comments could be qualified as „minor“, yet there are a couple of points that will probably require more attention from the authors. These „major“ comments are marked with an asterisk.    

  1. Lines 7-8: „However, previous research on these associations remains scarce and inconsistent. The present study examines these relationships in a sample of 695 fifteen-year-old students in Germany, differentiating between those with and without a migration background.“ – Please rephrase „these associations“ and „these relationships“ to be more specific and to establish more coherence with the previous sentence, in which the „association/relationship“ has already been narrowed down to an „influence“ of the first three variables (i.e., intelligence, academic achievement and underachievement) on the fourth one (i.e., SWB). In view of this, it seems that „these effects“ would be more suitable than „these associations“.

  1. Lines 19-21: „We discuss the implications for educational policy, emphasizing the need for targeted strategies to address underachievement among both migrant and native students.“ – I assume that we already know that this need exists, could you make this sound less generic and more related to your findings?

  1. Lines 27-32: „While subjective well-being (SWB) has been widely studied in relation to intelligence, academic performance, and educational outcomes, the strength of these associations remain debated. Some studies suggest that intelligence and academic achievement are positively linked to well-being (e.g. Bukhari & Khanam, 2017; Kirkcaldy et al., 2004), whereas others report only weak or inconsistent relationships (e.g. Gottfredson, 2008; Veenhoven & Choi, 2012). – Obviously, it is not simply the „strength of these associations“ that is being debated. From what you have written here, the question is also of their direction and whether there is an association in the first place.

  1. Lines 32-34: „Additionally, students who perform below their expected academic level—underachievers—may experience lower well-being (Van Batenburg-Eddes & Jolles, 2013), but this association is not yet fully understood. – Again, I find the ending of the sentence to sound rather generic. Certainly, based on prior research, there is more to be said about this association, than merely that it „is not yet fully understood“ (what is?). Please elaborate and provide more specific information.

  1. Lines 36-37: „Recent research also suggests that underachievement may be more prevalent among students with a migration background compared to native students...“ – Please provide a reference to back up this statement.

  1. Lines 38-40: „The present study aims to clarify these relationships by examining how intelligence, academic performance, and underachievement relate to SWB in students with and without a migration background.“ – Again, please specify „these associations“. Which ones? You are not equally interested in all possible associations between the given constructs, and you are not assigning the same status to all variables (some are treated as factors/predictors/independents, while some are treated as the outcome/criterion/dependent).

8.* Lines 40-42: „For the purposes of this study, we use the broad term subjective well-being (SWB) to encompass the more differentiated constructs of well-being, life satisfaction, and happiness (Diener, 1984; cf. Diener et al., 2009).“ – Indeed, SWB is a broad and complex construct, that needs to be presented in more detail, so that the reader can understand what is meant by „life satisfaction“ and „happiness“, and that they are not the same. In other words, the first requirement here would be to familiarize the reader with the difference between the various (i.e., „cognitive“ vs „affective“) aspects of SWB. The second requirement would be to heed these distinctions throughout the paper, especially when referring to inconsistencies in prior findings. For instance, your own study focuses on just one aspect of SWB – namely, life satisfaction – yet further on in the Introduction you present findings related to happiness, as a different aspect of SWB; all along, however, you treat both as representing SWB, not commenting on the possibility that intelligence and achievement might be differently related to life satisfaction vs happiness (see also my next comment).     

9.* Lines 56-58 and 74-76: „In their analysis of 23 studies taken from the World Database of Happiness, Veenhoven and Choi (2012) found no significant relationship between intelligence and happiness at the individual level, indicating that higher intelligence does not necessarily correspond to higher levels of happiness.“; „Some studies suggest that the relationship between intelligence and happiness is weaker among individuals with a migration background than among those without (e.g., Ali et al., 2013).“ – Please explicate the relation between the construct of „happiness“ and the construct of „life satisfaction“ and please mind the difference between them in your review of prior research. Drawing on one aspect of SWB or the other interchangably does not appear „fair“.

  1. Lines 76-77: „This may be attributed this to additional challenges and stressors that migrants face in the foreign country which may negatively impact their well-being (Veenhoven & Choi, 2012).“ – Delete the second „this“.

  1. Lines 78-80: „However, this difference in the relationships between intelligence and SWB between individuals with and without a migration background is not found consistently...“ – This appears to me as an unnecessary repetition, you have already stated in lines 73-74 that „previous studies remain inconclusive.“

  1. Lines 85-87: „Similar to the links between intelligence and SWB as detailed above, the links between academic achievement and SWB have also been widely studied and debated in the previous literature with varying perspectives on how these constructs influence each other.“ – This is probably debatable, but I would not say that „constructs“ can „influence each other“, because they are, well – constructs. It is the psychological qualities or variables behind the constructs that can be thought of as influencing each other.  

  1. Lines 108-109: „Previous studies relating underachievement to SWB supported these assumptions“ – It is not clear which assumptions. More precisely, the assumption is stated in lines 105-107, but there is a whole sentence (lines 107-108) disrupting the reference via a demonstrative pronoun (i.e., „these“ assumptions).

  1. Lines 116-117: „substantial differences between students with and without a migration background have been reported as well“ – It needs to be specified what these differences are (who is higher/lower on SWB).

  1. Line 125: „1.6 The Present Study“ – The number needs to be corrected to 1.3.

  1. Lines 126-128: „As detailed above, previous findings on the relations of intelligence, achievement, and underachievement with SWB are partially inconsistent and due to the scarce literature on the subject not well understood.“ – Again, I find this to be a rather vague and generic reflection on previous findings, using the same vocabulary that could do in almost any instance („inconsistent“, „scarce“, „not well understood“), but that does not really say much. Please avid such phrasings and be more informative and specific.  

  1. Lines 130-132: „The aim of the present study is to examine the relationships between intelligence, achievement, underachievement, and migration background in conjunction in a large sample of students at the age of 15.“ – I suppose that you were not really focused on the age of 15 as something specific, which is why I also believe that this age specification should not be part of your statement of the research aim. Otherwise, you should elaborate in the Introduction why you think that the associations between the chosen constructs deserve special attention at the age of fifteen.

18.* Lines 175-176: „General intelligence was assessed using the Culture Fair Test CFT 20-R by Weiß (2019), a widely applicable measure of fluid intelligence based on figural task material.“ – Please do not equate fluid intelligence with general intelligence. True, fluid intelligence usually has a very high loading on general intelligence, but general intelligence is necessarily a conglomerate of various cognitive abilities and cannot/must not be reduced to only one broad factor, be it Gf, Gc, or something else.   

19.* Lines 216-217: „To measure SWB, we used the seven-items Life Satisfaction subscale from the Habitual Subjective Well-Being Scale (Dalbert, 2002).“ – Nowhere in the paper is there an explanation why the construct (and assessment) of SWB was reduced to only one component (subscale). Please relate this to my comments No. 8 and 9 and look for a way to resolve them as one major issue.

  1. Line 223: „First, we provide the descriptive statistics and mean differences.“ – Referring to the differences, please state between whom.
  2. Lines 235-237: „Please note that grades here are used in their inverted scaling with lower scores indicating lower achievement and higher scores indicating higher achievement.“ – „Please note“ sounds a bit awkward here, perhaps you could rephrase with „Recall that grades...“ or use another wording.

  1. Lines 239-240: „Contrary to our first hypothesis, life satisfaction was unrelated to IQ in both groups“ – Though I admit that I am not a native speaker of English, I believe that „in either group“ would be more correct than „in both groups“.

  1. Lines 255-256: „In this case, the regression lines for underachievement and overachievement would need to diverge“ – The wording „need to diverge“ is a bit confusing, I would leave it at „would diverge“.

  1. Lines 274-275: „However, the difference between these correlation coefficients was just not significant“ – The phrase „was just not significant“ conveys something different from what I believe you meant to say. If I understand your intention correctly, you wanted to say that the correlation was statistically non-significant by a small margin. Please rephrase.

  1. Line 297: „First, our findings indicated that academic success was positively associated...“ – It is not the findings that indicated that, but the results of the statistical analyses. In other words, what you are describing here IS a finding. Please rephrase.

26.* Lines 300-307: „Differences in value systems and expectations could account for this pattern. There is a correlation between the level of family socio-economic status and educational attainment of young people. Because students with a migration background are more likely to be in low-income home environments (Blom & Severiens, 2008), they are at particular risk academically. Academic success may therefore be more closely linked to life satisfaction for students without a migration background, as their families and social environments could emphasize academic achievement as a key determinant to personal fulfilment.“ – In my view, the middle part of the quoted text is completely superfluous and interrputs the logical flow of the narrative. The section would read better as follows: „Differences in value systems and expectations could account for this pattern. Specifically, academic success may be more closely linked to life satisfaction for students without a migration background, as their families and social environments could emphasize academic achievement as a key determinant to personal fulfilment“ (while the families of students with a migration background might convey other values as more important). Please consider a revision here.

27.* Lines 318-319: „As noted in prior research, numerous studies have failed to establish a positive relationship between these two constructs“ – If there are indeed numerous studis that have failed to establish a positive relationship between the given constructs, then you should seriously reconsider your hypothesis.

28.* Lines 320-321: „Veenhoven and Choi (2012) demonstrated that individuals with higher IQs do not necessarily experience greater happiness“ – Here, again, „life satisfaction“ is simply substituted by „happiness“, though they have been theoretically and empirically distinguished as two different aspects of SWB.

29.* Lines 327-330: „However, this assumption was not supported by our findings. Moreover, students with a migration background reported higher life satisfaction than those without, a result that contrasts with many previous studies (Marquez & Long, 2021; Ullmann & Tatar, 2001; Vieno et al., 2009). – You begin here by discussing the relationship between life satisfaction and underachievement, but then you suddenly jumpt to the differences in life satisfaction between the two groups in your study. Please follow a clear line of thought and do not unnecessarily diverge from the topic.

  1. Lines 341-342: „The perception of not being a minority, despite being classified as having a migration background, could explain our findings.“ – Please specify which of your findings.

  1. Lines 356-357: „As it relies on cross-sectional data, the extent to which the findings can be generalized at an individual level remains unclear.“ – I am not sure that I understand this statement, a generalization should by definition go from something particular to something broader, not the other way around (arriving at an individual level). Please revise.

Reviewer 3 Report

Comments and Suggestions for Authors

Thank you for the opportunity to review the paper entitled “Understanding How Intelligence and Academic Underachievement Relate to Life Satisfaction Among Adolescents With and Without a Migration Background”

It is an interesting paper and provide some important implications, especially focusing on the role of migration and intelligence in wellbeing.

With the intro, it is important to conceptualize each construct and provide a theoretical framework. For example, how you conceptualize intelligence or SWB? Why do you expect intelligence can be associated with SWB?

Please provide more info about the sample, such as age range.

Why did you just use life satisfaction from the measure for wellbeing? because the scale has an emotional dimension.

Please provide info about all measures with psychometric in the culture

It is important to present data analyses section because it is difficult to understand why some analyses were conducted.

How was Culture Fair Test CFT 20-R applied?

The study interestingly  indicated that students with a migration background reported higher life satisfaction than those without a migration background. Also, there is no a correlation between Intelligence  and life satisfaction, regardless of migration background. The author should discuss this result and provide a deeper understanding.

Intelligence may be moderate the effect immigration on wellbeing.

Also, theoretical and practical implications should be provided for researcher and practitioners.  

Round 2

Reviewer 1 Report

Comments and Suggestions for Authors

Congratulations to the authors for their work.
I recommend the publication of the paper in its present form.

Reviewer 2 Report

Comments and Suggestions for Authors

Thank you for including me in the evaluation of the revised version of this paper. I also thank the authors for carefully considering my comments and suggestions given in the first round of reviews. Overall, I find that the authors have done a good job of revising their original manuscript and I only have a couple of minor (in many cases, stylistic) comments on the text preceding the Discussion (see below).

I do have the impression, however, that the Discussion does not match the rest of the text in terms of richness and length, and I would kindly ask the authors to revise and expand this section, so as to revisit all their research hypotheses as stated in „The Present Study“ (i.e., the structure of the Discussion should mirror the list of hypotheses). That would be my only major request.     

  1. Lines 5-6: „Intelligence, academic achievement, and an unfavorable discrepancy between intelligence and achievement (i.e., underachievement)...“ – Perhaps you could avoid the repetition and shorten this to: „and an unfavorable discrepancy between them“

  1. Line 8: „The present study examines these correlations in a sample...“ – Again, „these correlations“ does not have a clear syntactic reference (which correlations?). I would suggest rephrasing as follows (or similarly): „The present study examined the associations between the said variables...“   

  1. Lines 8 – 12: „The present study examines these correlations in a sample of 695 fifteen-year-old students in Germany, differentiating between those with and without a migration background. Our findings unexpectedly reveal that students with a migration background reported higher life satisfaction than those without a migration background. Intelligence was unrelated to life satisfaction, regardless of migration background“ – The past tense should be used consistently. Thus „examines“ should be changed to „examined“, and „reveal“ to „revealed“.

  1. Lines 34-35: „not only the strength but even the direction of these presumed associations remain unclear“ – It should be „remains“ not „remain“, because the verb refers to „direction“ and „strength“ (singular)

  1. Lines 70-71: „On the one hand, there is fluid intelligence, which is closely linked to cognitive processing...“ – All cognitive abilities involve cognitive processing, so the definition of Gf should be more specific and refer to the ability to solve novel problems based on logical reasoning (not relying on acquired knowledge).

  1. Lines 94-95: „Due to recent societal changes in many countries due to increasing rates of international migration...“ – Please avoid using „due to“ twice here. Maybe say: „Due to recent societal changes in many countries, involving increasing rates of international migration...“

  1. Lines 115-116: „...with varying perspectives on how the variables behind those constructs influence each other..“ – There is no need for this complicated phrase, just „on how these variables influence each other“ is clear enough.

  1. Lines 175-177: „Differences in these relations between students with and without a migration background will be conducted exploratorily..“ – I do not think that you can „conduct differences“, so this should be rephrased. One possibility would be to say: „Differences in these relations... were treated exloratorily, without a specific hypothesis being put forth.“ Please use past tense when reporting about a study that you have already conducted.

  1. Lines 190-193: „723 participants took part in the study. Of these, 28 were excluded due to highly implausible response patterns such as consistently selecting only the maximum or the minimum response option across all items of the questionnaire regardless of inverted items. These patterns suggested a lack of honest participation.“ – Given that this is just contextual data, I would suggest summing this up in a single sentence, as follows: „Of these, 28 were excluded due to highly implausible response patterns, suggesting a lack of honest participation (e.g., consistently selecting only the maximum or the minimum response option across all items of the questionnaire regardless of inverted items)“

  1. Lines 387-388: „As mentioned at the beginning, there have been some studies that have not been able to report a positive relationship between intelligence and life satisfaction“ – Please rephrase „not been able to report“ to „have not found“ or smt. similar (because it is not ability or inability that is the reason here).

Round 3

Reviewer 2 Report

Comments and Suggestions for Authors

I have no further suggestions.